# Reconstruction of the Vertical Dynamic Running Load from the Registered Body Motion

Katrien Van Nimmen [1,*], Benedicte Vanwanseele [2] and Peter Van den Broeck [1]

1 Department of Civil Engineering, Structural Mechanics, KU Leuven, B-3001 Leuven, Belgium
2 Department of Movement Sciences, Human Movement Biomechanics, KU Leuven, B-3001 Leuven, Belgium
* Correspondence: katrien.vannimmen@kuleuven.be

**Abstract:** In view of in-field applications, this paper introduces a methodology that uses the registered body motion to reconstruct the vertical dynamic running load. The principle of the reconstruction methodology is to use the time-variant pacing rate that is identified from the body motion together with a generalized single-step load model available in the literature. The methodology is reasonably robust against measurement noise. The performance of the methodology is evaluated by application to an experimental dataset where the running load and the body motion were registered simultaneously. The results show that a very good fit is found with the measured forces, with coefficients of determination of 95% in the time domain and 98% for the amplitude spectrum. Considering a 90% confidence interval, the fundamental harmonic is shown to be reconstructed with a maximum error of 12%. With nearly 90% of the energy concentrated around the fundamental harmonic, this harmonic is the dominant component of the running load. Due to the large inter-person variability in the single-step load pattern, a generalized single-step load model does not arrive at a good fit for the higher harmonics: the reproduction errors easily exceed 50% for a 90% confidence interval. Finally, the methodology is applied to reproduce the dynamic running load induced during full-scale tests on a flexible footbridge. The tests are designed such that the structural response is governed by the (near-)resonant contribution of the fundamental harmonic of the running load. The results show that even when a 12% uncertainty bound is taken into account, the structural response is significantly over-estimated by the numerical simulations (up to 50%). These results suggest a non-negligible impact of other phenomena, such as human–structure interaction, that are not accounted for in current load models.

**Keywords:** vibration serviceability; human-induced vibrations; footbridge; running



## 1. Introduction

Vibration serviceability under loading induced by human actions is a key design criterion for footbridges [1,2]. Over the last two decades, the dynamic performance of footbridges under loads induced by walking has been widely investigated [1,3–6]. However, little to no attention has been paid to running actions as a design load. The concerns about the impact of dynamic running actions (single or in group) have grown significantly as (1) the involved load amplitudes are significantly higher than for walking, and thus, additional structural modes may be critical for the vibration serviceability, (2) the dominant loading frequencies are higher than for walking and (3) footbridges are progressively more exposed to running due to the increased focus on a healthy lifestyle. The latter also explains the growing popularity of marathons and urban trails.

As running-induced loading is predominantly vertical, from an civil engineering point of view, focus is nearly always on its vertical component. This is also the case for the present study. The state of the art involving dynamic running actions is limited to single-person load models valid for running actions on a rigid laboratory floor [7,8]. Such load models were recently used by Occhiuzzi et al. [9] to investigate the effectiveness of (semi-active)

tuned mass dampers (TMDs) for the control of running-induced vibrations of footbridges. Other recent contributions focus on the further development of single-person load models based on single-person tests performed on a rigid laboratory floor. Based on a total of 458 force–time histories (generated by 45 participants) containing at least 64 successive footfalls, Racić and Morin [10] developed a data-driven mathematical model designed to generate random signals representative of the intra-person variabilities (both temporal and spectral features) as observed in the measured forces. As this model is designed to generate random samples of a random process, it can be used in design calculations, but it is not suited to replicate specific experimental events. More recently, Pańtak [11,12] introduced load models for heel-strike and forefoot-strike running based on a large set of single-step load traces measured for 13 participants for pacing rates between 2.40 and 3.40 Hz. Other advances can be found in the biomechanical community, for example [13–15], where the focus is on the running kinematics, mechanics and the resulting ground reaction forces (GRFs). In this community, single-person tests performed on a rigid laboratory floor are standard practice.

So far, however, no expertise is available on running excitation as a load scenario for civil engineering structures. The load model presented by currently available standards and design guidelines [16,17] represents a single or a small group of runners as perfectly periodic and perfectly synchronized individuals. The structural response is then calculated assuming resonant conditions with each of the relevant structural modes. These completely unwarranted assumptions lead to unrealistically high predicted vibration levels that compromise any slender structural design or require the installation of expensive vibration mitigation devices. The development of realistic load scenarios for running excitation requires input on:

- Human–Human Interaction (HHI): When individuals walk or run in small/large groups or large crowds, their locomotion is influenced by that of neighboring individuals [18–20]. For walking excitation, this is known to influence the synchronization rate [21]. Design guides therefore specify a distribution of step frequencies as a function of the pedestrian density [16,17]. Because the vibration serviceability assessment is in nearly all cases conducted under resonant conditions, the synchronization rate among individuals has a normative influence on the structural design.
- Human–Structure Interaction (HSI): This phenomenon involves the so-called (1) active HSI where the locomotion is influenced by the motion of the supporting surface (footbridge) and (2) passive HSI that involves the mechanical interaction between the human body and the supporting structure. For walking excitation, active HSI phenomena have been shown to considerably influence the level of synchronization rate as well as the load amplitudes, which are two features that have a significant impact on the resulting structural response [3,5,22]. In addition, passive HSI has been shown to be normative for the VSA design, in particular for larger groups and crowds, where the dominant effect is added damping [23–25].

The investigation of HHI and (passive) HSI requires in-field observations and, generally, a large number of participants. In these cases, direct force measurements are practically infeasible. An interesting alternative is the use of indirect force measurements, where the induced loads are reconstructed from the registered body motion. Recent contributions have successfully reconstructed the ground reaction forces (GRFs) induced by jumping, bobbing and walking using either visual markers [26–29] or inertial sensors [30–35]. In laboratory conditions, the golden standard uses marker-based technologies. Due to issues with occlusions and the distance limitations in terms of tracking, this technique is infeasible for in-field applications involving a large number of persons. In answer to these limitations, in-field applications have to resort to individually adopted sensor technologies, for example inertial sensors [30–35].

The present study focuses on in-field applications. A methodology is presented to reconstruct the vertical running load using the registered body motion and a literature load model as inputs. The methodology is based on the one recently introduced for

the reproduction of the vertical walking load [36]. The methodology is designed to be reasonably robust against measurement noise arising from the use of low-cost sensor technologies and soft-tissue artifacts. The body motion only needs to be registered at a single location on the human body and only requires a relatively low sampling rate (e.g., 25 Hz). The registered body motion is then used to identify the time-variant pacing rate. In doing so, the methodology does not rely on the position of the sensor on the human body nor the exact magnitude of the registered body motion. The identified time-variant pacing rate is then combined with a literature single-step load model to reconstruct the induced running load [36]. In doing so, the methodology does not account for small step-by-step variations in load amplitudes or contact time. However, as the structural response is predominantly sensitive to variations of the pacing rate (and the weight) [30,37], this simplification can be reasonably accepted for applications in structural dynamics.

The procedures used in the experimental work are approved by the social and societal ethics committee of KU Leuven, and each subject gave written informed consent prior to participation.

The outline of this paper is as follows. In Section 2, the reconstruction methodology is discussed. In Section 3, the performance of the reconstruction methodology is validated using laboratory experiments and is then applied on full-scale tests on a flexible footbridge. Section 4 summarizes the conclusions.

## 2. Reconstruction Methodology

A requirement for the reconstruction methodology is to be suitable for application to in-field experiments involving multiple pedestrians. The feasibility of this type of experiment relies on the use of wireless low-cost sensors. In comparison to the (marker-based) techniques that serve as a golden standard in laboratory conditions, these low-cost sensors are associated with higher levels of low-frequency drift, lower sample rates and lower sensitivities. In addition, as the targeted in-field experiments involve the application of the sensors on top of (multiple layers of) clothing, the impact of soft-tissue artifacts [28,38] is expected to be larger than what is classically observed in the laboratory. It is for these reasons that the reconstruction of the human-induced load using the data collected by these sensors requires a carefully conceived approach.

For reference purposes, Section 2.1 presents a first reconstruction methodology that is defined in line with Newton's second law (Section 2.1): the external force acting upon a body equals the product of its mass concentrated at the body center of mass (BCoM) and its acceleration. In doing so, there is a direct relation between the accuracy of the reproduction and the accuracy of the sensor used to register the body motion. Additional error sources for this reconstruction method include soft-tissue artefacts and the assumption that the kinematics as the BCoM are identical to those of a single point on the body surface [26].

In Section 2.2, an alternative methodology is proposed. The principle of this alternative method is to reconstruct the running load from the time-variant pacing rate, identified from the registered body motion, and a generalized single-step load model available in the literature [39,40]. This methodology is based on the one recently introduced for the reconstruction of walking-induced loads [36]. In this case, the accuracy of the reconstructed running load does not depend on the exact magnitude of the registered body motion but on the accuracy of the identified time-variant pacing rate. The methodology only relies on capturing the body motion within the frequency range 1–10 Hz. For the case of walking, the methodology is shown to be suitable for application to low-cost sensors and to be reasonably robust against soft-tissue artifacts [36].

### 2.1. Newton's Second Law

Theoretically, the GRFs during locomotion can be calculated from Newton's second law that states that the external force acting upon a body equals the product of its mass

$m$ concentrated at the body center of mass (BCoM) and its acceleration $\ddot{u}_{\mathrm{COM}}$. The vertical GRFs acting on the human body ($p_{\mathrm{GRF}}$) then read:

$$p_{\mathrm{GRF}}(t) = m\big(\ddot{u}_{\mathrm{COM}}(t) + g\big) \tag{1}$$

where $g$ represents gravity and $G = mg$ [N] represents the weight of the body. Alternatively to Equation (1), biomechanical inverse dynamic analyses quantify the GRFs by combining kinematic measurements on $n_s$ segments of the human body with the segment's inertial parameters (mass and BCoM) [41,42]. This approach comes with its own challenges and is not feasible for the present application, which is primarily because it requires each individual to be equipped with a large set of sensors.

The reconstruction of the GRFs using Equation (1) uses the nominal body mass $m$ and the acceleration measured close to the BCoM ($\ddot{u}_{\mathrm{COM}}(t)$). For this purpose, often, the location at the level of the 5th lumbar vertebrae is selected. The main sources for the errors are soft-tissue artefacts and the assumption that the kinematics as the BCoM are identical to those of a single point on the body surface [26]. The BCoM is not directly accessible, as it is located within the person's trunk [26]. Furthermore, the location of the BCoM is not fixed throughout the running cycle. The changes in the location of the BCoM are more pronounced for running than for walking, as the more energetic body motion is characterized by larger relative motions of the individual body segments. For walking, the error related to the BCoM is often reduced by applying magnitude scaling $\alpha_{\mathrm{eff}}$ to the body mass. This then results in an effective body mass $m_{\mathrm{eff}} = \alpha_{\mathrm{eff}} m$ [43]. The average scaling factor ($\bar{\alpha}_{\mathrm{eff}}$) can be estimated as the linear magnitude scaling factor that minimizes the difference between the power spectral density (PSD) of the reconstructed GRFs and the PSD of the measured GRFs [36]. The optimal scaling factor $\alpha_{\mathrm{eff}}$ depends on the participant and the walking speed, and it displays large variations between 42% and 92% [36,44,45]. It is shown that for walking, the reconstruction of the GRFs using Newton's second law is fair. That being so, this reconstruction methodology has two drawbacks. First, using single-point kinematics to reconstruct the GRFs implies the proneness of the methodology to soft-tissue impact dynamics. The optimal scaling factor $\alpha_{\mathrm{eff}}$ not only displays large inter-person variability but is also influenced by the locomotion's intensity and speed. To overcome this issue, reference tests should be performed in laboratory conditions for each participant that is involved in in-field tests. This is, however, not feasible for in-field tests involving a large number of participants. Second, this reconstruction methodology requires acceleration levels registered close to the BCoM. Additional reconstruction errors may result from changes in the orientation or location of the sensors.

### 2.2. Using the Time-Variant Pacing Rate and a Generalized Load Model

2.2.1. Identification of the Time-Variant Pacing Rate

The time between two successive steps is defined as the time between two nominally identical events of two successive cycles in the human locomotion [46]. The pacing rate is the inverse of the time between two successive steps. The time-variant pacing rate is then defined as the collection of pacing rates in time identified for a series of steps. To identify the time-variant pacing rate from the registered body motion with a high level of accuracy (error $\leq 1\%$), the method introduced in [36] is used and adapted for application to running actions. The first step consists of preprocessing the registered body motion:

- The data are resampled at 1000 Hz: Given the accuracy that is aimed for when identifying the time-variant pacing rate (1%), it is recommended to use a resolution in the time domain of 0.001 s. For the running cycle, this resolution corresponds to 0.3% of the smallest period that reasonably can be expected, i.e., corresponding to a pacing rate of 3.5 Hz;

- The PSD of the registered body motion is calculated with a frequency resolution of at least 0.05 Hz. The mean pacing rate $\bar{f}_p$ is identified as the dominant contribution of the PSD in the relevant frequency range: 2.0–3.5 Hz.

- The data are low-pass filtered with a cut-off frequency at $1.5\bar{f}_p$.

The second step consists of identifying the time-variant pacing rate. To do so, the timing of the subsequent load cycles is identified: that is, the timing of the peaks of the filtered signal. The actual onset of each step ($t_{o,k}$) is found by applying the time shift that maximizes the correlation between the reconstructed GRFs and the vertical accelerations registered near the BCoM.

### 2.2.2. Running Load Models

This study evaluates the performance of four running load models reported in the literature (see Table 1).

**Table 1.** The single-step load patterns used in this study.

| Name | ID | Type | Description |
|---|---|---|---|
| Reference model | ref | measured | load pattern per participant per trial |
| Half-cycle sine pulses | 1 | generalized | cfr. Wheeler [7], Bachmann & Ammann [8] |
| Heel-strike load pattern | 2 | generalized | cfr. Pańtak in [11,12] |
| Forefoot-strike load pattern | 3 | generalized | cfr. Pańtak in [11,12] |

For reference purposes, a first running load model is determined as the average single-step load pattern of the participant for the involved treadmill trial. When this load model is used, the reconstruction error reflects the impact of the error on the identified time-variant pacing rate as well as the step-by-step variations in the load pattern that are not taken into account in the applied reconstruction method.

The second running load model that is used is the widely-applied load model, as introduced by Wheeler [7] and Bachmann and Ammann [8]. This load model specifies a sequence of half-cycle sine pulses $p_1(t)$:

$$p_1(t) = \begin{cases} k_p G \sin\left(\frac{\pi t}{t_c}\right) & 0 \leq t \leq t_c \\ 0 & t_c < t \leq t_p \end{cases} \tag{2}$$

where $k_p$ is the dynamic impact factor, $G$ is the weight of the person, $t_c$ is the contact duration and $t_p$ is the pace period of the activity, i.e., the inverse of the pacing rate $f_p$ [47]. The intensity of the activity is inversely proportional to the contact duration ratio $t_c/t_p$ and proportional to the forcing amplitudes. The contact duration $t_c$ follows from Wheeler's relation [7]:

$$t_c = -0.0724 + \frac{0.5463}{f_p - 0.9078} \tag{3}$$

The impact factor $k_p$ then results from the condition of constant potential energy [8]:

$$k_p = \frac{\pi}{2\frac{t_c}{t_p}} \tag{4}$$

The load parameters defined by Wheeler [7], Bachmann and Ammann [8] are the result of a comprehensive systematization of the (experimental) work of other researchers. Although these parameters are defined to represent the average load induced by running, it is noted that they (may) differ among individuals.

Third, the load models recently introduced by Pańtak [11,12] are used. His results are based on a large set of single-step load traces measured for 13 participants for pacing rates between 2.40 and 3.40 Hz (frequency increment of 0.2 Hz). Three load patterns are distinguished:

- Heel-strike running: characterized by an impact peak (a first small peak) and a propulsive peak (a second big peak, also known as the active peak when the body center of mass moves over the foot);
- Forefoot-strike running: only one active peak occurs;
- Midfoot-strike running: similar to forefoot-strike running with only a slight disturbance near the position of the impact peak.

In [11,12], Pańtak introduces load models for forefoot-strike and heel-strike running. Both models are used in this work. Similar to Bachmann and Ammann [8], the forefoot-strike load pattern is a half-cycle sine pulse. The difference lies in the definition of the impact factor and the contact ratio. Similar to the proposal of Racić and Morin [10], the heel-strike load pattern is modeled as the sum of five Gaussian functions. For both load patterns, the required input parameters are defined in terms of the pacing rate. For reasons of conciseness, the reader is referred to Pańtak's [11,12] work for an overview of these load parameters.

Table 1 gives an overview of the different load models used in this work. Figure 1 shows the single-step load traces generated by the three generalized load models for two pacing rates: 2.5 Hz and 3.0 Hz. This figure shows that overall, the models are generally consistent. Small differences can be observed for the peak amplitude and contact time, in particular as the pacing rate increases.

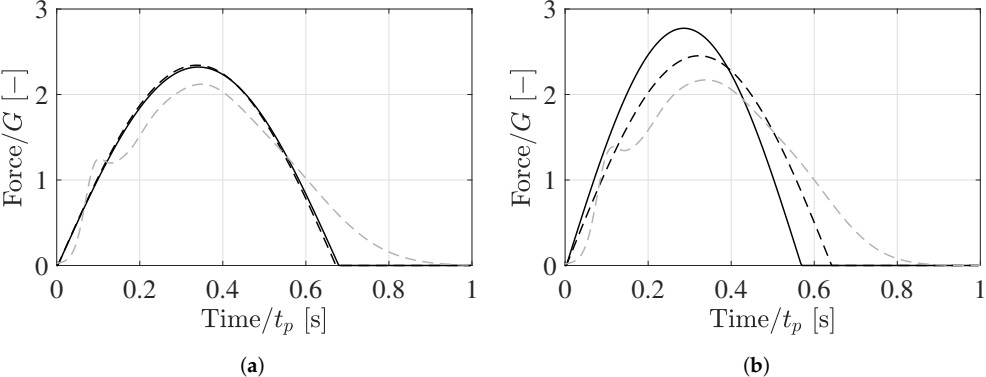

|     (a)     |     (b)     |

**Figure 1.** Single-step load pattern according to Wheeler [7], Bachmann and Ammann [8] (black, solid), Pańtak [11,12] (forefoot-strike, dashed, black), and Pańtak [11,12] (heel-strike, dashed, gray) for a pacing rate $f_p = 1/t_p$ of (**a**) 2.5 Hz and (**b**) 3.0 Hz.

2.2.3. Reconstruction

The vertical GRFs are then reconstructed as follows. The force due to a single step $k$ reads:

$$p_k(t) = \kappa(t - t_{o,k})\bar{p}_1(t - t_{o,k}), \qquad \text{with} \quad \kappa(t) = \begin{cases} 1 & 0 \leq t \leq \bar{t}_c \\ 0 & \text{otherwise} \end{cases} \tag{5}$$

with $t$ [s] being the general time of the force signal, $\bar{p}_1$ being the generalized single-step load pattern, onset $t_{o,k}$ being the onset of step $k$, and $\bar{t}_c$ [s] being the generalized duration of the contact between the foot and the ground. The reconstructed GRFs then read:

$$p_{\text{GRF}}(t) = \sum_k p_k(t) \tag{6}$$

*2.3. Evaluation Methodology*

To evaluate the overall fit between the original and the reconstructed forces, this study uses the coefficient of determination $R^2$ [26,48] for both the time series ($R_t^2$) and the amplitude spectrum ($R_f^2$). In addition, the reconstruction quality is evaluated per harmonic component of the running load. To do so, the following formulation of the GRFs is considered:

$$p_{\mathrm{GRF}}(t) = G + G \sum_{h=1}^{n_h} p_h(t) \tag{7}$$

with:

$$p_h(t) = \sum_{f=f_{lh}}^{f=f_{uh}} \mathrm{A}_h(f) \cos(2\pi f t + \theta(f)) \tag{8}$$

with $n_h$ being the number of harmonics $h$, $f = [f_{lh}, f_{uh}]$ being the frequency range of the harmonic (with $0.4 f_p < f_{lh} < f_p$ and $f_p < f_{uh} < 1.4 f_p$), and $\mathrm{A}_h(f)$ and $\theta(f)$ being the amplitude and phase of the corresponding line in the spectrum, respectively. The reconstruction quality per harmonic is then evaluated by the ratio $R_h$ [-]:

$$R_h = \frac{\sum_{f=f_{lh}}^{f=f_{uh}} \mathrm{A}_h(f)}{\sum_{f=f_{lh}}^{f=f_{uh}} \tilde{\mathrm{A}}_h(f)} \tag{9}$$

with $\mathrm{A}_h(f)$ and $\tilde{\mathrm{A}}_h(f)$ being the amplitudes of the corresponding lines in the spectrum of the true force and the reconstructed force, respectively. $R_h$ being lower than unity means that the reconstructed forces overestimate the true forces and vice versa. Unless otherwise specified, this study considers the interval $[f_{lh} = (h - 0.4)\bar{f}_p, f_{uh} = (h + 0.4)\bar{f}_p]$ to evaluate $R_h$.

## 3. Experimental Validation

### 3.1. Laboratory Experiments

This study uses data collected at the Movement & posture Analysis Laboratory Leuven (MALL) at the Department of Movement Sciences of KU Leuven [49]. To allow for the investigation of stride-to-stride variations [4], treadmill data are used.

In total, 13 experienced treadmill users took part in this study (Table 2). The GRFs were registered by an instrumented split-belt treadmill (Forcelink) with a sampling rate of 1 kHz. Prior to the trials used in this study, the participants ran on the treadmill at a self-selected speed during 5 min to familiarize with the treadmill running activity. Then, six running speeds were considered for each participant (see Table 2). These speeds were chosen in the range that was considered comfortable for the involved participant. For each test, a specific speed was selected and an initiation period (approximately 30 s) was provided allowing the participant to adjust to the treadmill speed. Next, data were collected during 2 min. In this way, approximately 300 steps were registered for each test. Rests were provided between successive tests. The body motion was registered using a wireless inertial unit (type GCDC X-16D [50]) at 200 Hz. This unit was securely fixed near the 5th lumbar vertebrae (L5). On top of this wireless inertial unit, a visual maker was placed. The motion of this visual marker was measured using a 10-camera Vicon motion capture system [51] with a sampling frequency of 200 Hz.

The collected data are preprocessed as follows. The force plate data are decimated to 200 Hz. All signals are low-pass filtered with a cut-off frequency of 20 Hz. By doing so, the impact of measurement noise is minimized and the frequency content relevant for the vibration serviceability assessment of footbridges (0–10 Hz, scope of application) is retained. The visual marker and force plate data are synchronously registered. These data are then synchronized offline with the data collected by the inertial units. To do so, the cross-correlation between the accelerations measured directly by the inertial units and the accelerations derived from the marker data is maximized. The remainder of this paper only uses the (synchronized) accelerations registered by the inertial units.

**Table 2.** Participant information (Male (M), Female (F)).

| Participant | Sex | Age | Length (m) | m (kg) | Running Speeds (km/h) |
|---|---|---|---|---|---|
| 1 | M | 25 | 1.82 | 77 | (9; 9.5; 10; 10.5; 11; 11.5) |
| 2 | M | 22 | 1.77 | 72 | (7.5; 8; 8.5; 9; 9.5; 10) |
| 3 | M | 51 | 1.75 | 77 | (9; 9.5; 10; 10.5; 11; 11.5) |
| 4 | M | 27 | 1.89 | 72 | (10; 10.5; 11; 11.5; 12; 12.5) |
| 5 | M | 28 | 1.90 | 85 | (10; 10.5; 11; 11.5; 12; 12.5) |
| 6 | F | 32 | 1.66 | 54 | (7.5; 8; 8.5; 9; 9.5; 10) |
| 7 | M | 24 | 1.87 | 82 | (9; 9.5; 10; 10.5; 11; 11.5) |
| 8 | M | 24 | 1.85 | 79 | (9.5; 10; 11; 12; 12.5; 13) |
| 9 | M | 24 | 1.73 | 71 | (9.5; 10; 10.5; 11; 11.5; 12) |
| 10 | M | 21 | 1.85 | 78 | (9; 9.5; 10; 10.5; 11; 8.5) |
| 11 | M | 22 | 1.85 | 77 | (8; 8.5; 9; 9.5; 10.5; 12) |
| 12 | M | 22 | 1.65 | 74 | (10; 10.5; 11; 11.5; 12; 12.5) |
| 13 | M | 21 | 1.77 | 81 | (9; 9.5; 10; 10.5; 11.5; 12) |

*3.2. Reconstruction of the Running Load*

First, focus is on the energy per harmonic and the averaged single-step load pattern. Next, the performance of the reconstruction methods is evaluated.

3.2.1. Energy Per Harmonic

As for walking-induced loading, running-induced loading is narrow-band (and therefore 'imperfectly' periodic), and its energy is concentrated around the lowest harmonics. The fundamental harmonic ($h = 1$) is defined as the harmonic with a center frequency equal to the average pacing rate $\bar{f}_p$. The higher harmonics ($h \geq 2$) have their center frequency at $h \times \bar{f}_p$ and an integer multiple of $\bar{f}_p$. The narrow-band nature of the running load is illustrated in Figure 2a.

The energy $E_h$ per harmonic $h$ is defined as:

$$E_h = \int_{(h-0.4)\bar{f}_p}^{(h+0.4)\bar{f}_p} |X(f)|^2 df \tag{10}$$

with $X(f)$ being the Fourier transform of the running load as a function of frequency $f$. For each of the tests, the energy per harmonic was calculated and normalized to the energy concentrated around the fundamental harmonic ($E_1$). Figure 2b shows that the energy concentrated around the second, third and fourth harmonic corresponds to respectively 8.6% (95-percentile interval: $\pm 6.1\%$), 2.7% (95-percentile interval: $\pm 1.9\%$) and 1.9% (95-percentile interval: $\pm 1.5\%$) of the energy concentrated around the fundamental harmonic. The energy concentrated in the higher harmonics thus only represents a fraction of the energy concentrated in the fundamental harmonic.

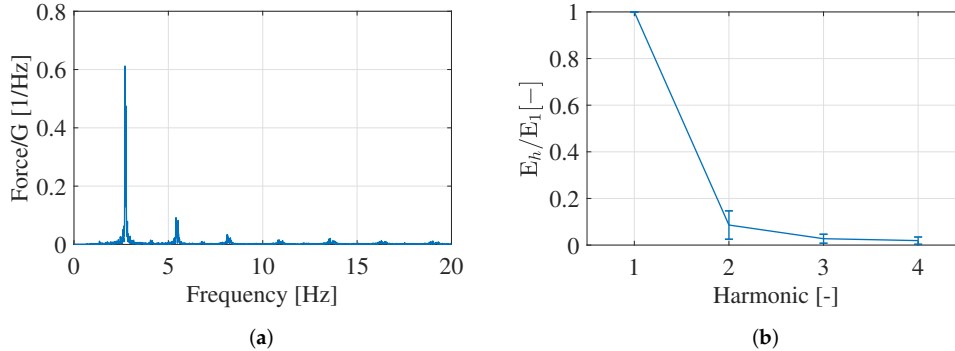

(a)　　　　　　　　　　　　　　(b)

**Figure 2.** (**a**) The amplitude spectrum of the load induced by participant 1 during test 1; (**b**) The energy per harmonic ($E_h$), normalized to $E_1$, for all laboratory tests, the error bar represents the 95-percentile of the samples.

### 3.2.2. Averaged Single-Step Load Pattern

The reconstruction of the running load according to Section 2.2 requires a generalized single-step load pattern. For reference purposes, the average single-step load pattern of the participant for the involved treadmill trial is used. This average single-step load pattern is obtained by aligning the individual single-step load patterns by their peak value and then by taking the average at each time step. Figure 3 illustrates the averaged single-step load pattern for participant 1 and test 1.

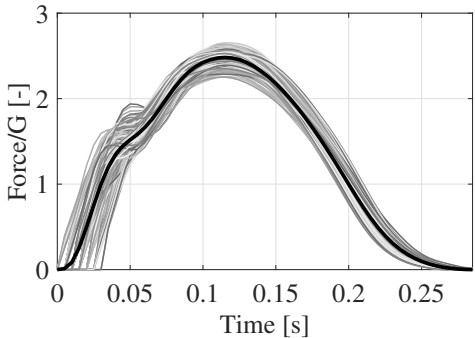

**Figure 3.** For participant 1 test 1: All identified single-step load patterns (gray) and the average single-step load pattern (black, thick line).

### 3.2.3. Performance of the Reconstruction Methods

First, focus is on the performance of the reconstruction using Newton's second law (Section 2.1). Figure 4 compares the time series and amplitude spectrum of the measured and the reconstructed forces for participant 1 test 1. This figure shows that, as expected, the reconstructed forces overestimate the measured ones. In addition, Figure 4b shows that in comparison to the measured forces, more energy is situated in the higher and sub-harmonics of the reconstructed forces. This indicates that the presence of these (sub-)harmonics is more pronounced in the body motion than in the resulting forces. These observations are confirmed by those obtained for all tests: an average magnitude scaling factor $\alpha_{\text{eff}}$ of 61% is found for the amplitude spectrum (0–10 Hz), with 25- ($P_{25}$) and 75-percentile values ($P_{75}$) of respectively 56% and 67%. A 50-percentile value ($P_{50}$) equal to 61% means that the reproduction using Newton's second law, on average, overestimates the magnitude of the true forces by approximately 40%. Concerning the reconstruction quality per harmonic, an average $R_h$ of 80% ($h = 1$, $P_{25} = 75\%$, $P_{75} = 85\%$), 63% ($h = 2$, $P_{25} = 42\%$, $P_{75} = 83\%$), 34% ($h = 3$) and 28% ($h = 4$) is found (see Figure 5a). In time and frequency domain, an average coefficient of determination of respectively 52% ($P_{25} = 36\%$, $P_{75} = 70\%$) and 83% ($P_{25} = 78\%$, $P_{75} = 91\%$) is found. These results show that this method is not suited to reconstruct the running load for $h \geq 2$ and that the reconstruction quality for the fundamental harmonic is, as best, fair. Furthermore, the coefficient of determination obtained for the time domain is low.

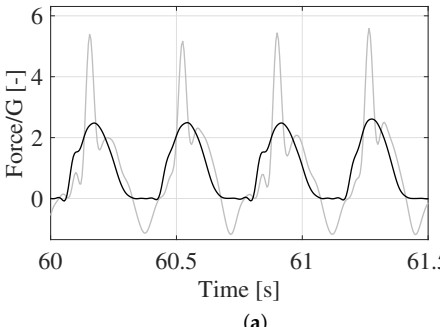
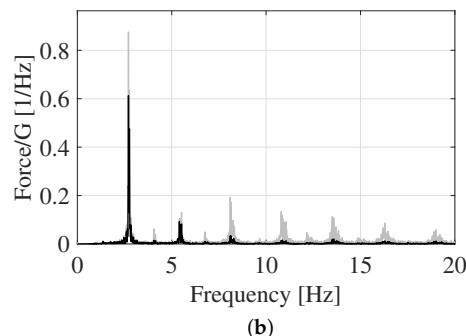

(**a**)  (**b**)

**Figure 4.** The load induced by participant 1 during test 1, measured forces (black), reconstructed using Newton's second law (gray): (**a**) time series, (**b**) amplitude spectrum.

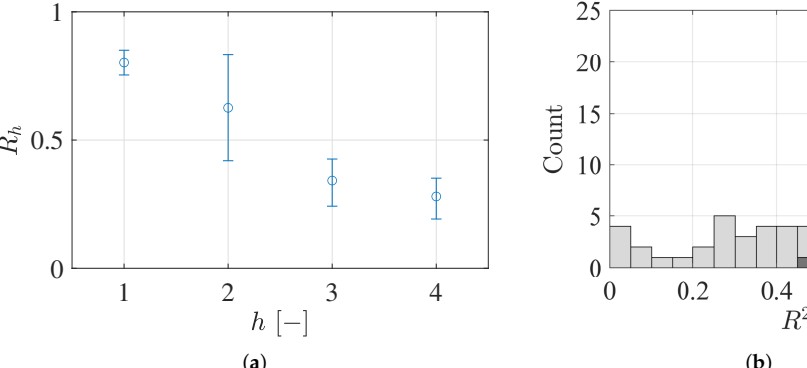

**Figure 5.** (**a**) $R_h$ for the first four harmonics: mean value (○) and error bar indicating the ($P_{25} - P_{75}$) interval; (**b**) Histogram of the coefficients of determination for the time domain ($R_t^2$, light) and the frequency domain ($R_f^2$, dark).

Next, focus is on the performance of the reconstruction method introduced in Section 2.2. This construction method requires the time-variant pacing rate and a generalized single-step load trace as input. Section 2.2.1 specifies how the time-variant pacing rate is determined. Concerning the single-step load pattern, this study investigates the performance of four different models (see Table 1). The first load pattern (Model 'ref' in Table 1), the averaged single-step load pattern of the participant for the involved treadmill trial, is applied for reference purposes. Reconstruction errors that are observed when using this model arise from disregarding the step-by-step variations in the single-step load pattern or errors in the identified time-variant pacing rate.

The second single-step load pattern (Equation (2), Model 1 in Table 1) requires the weight of the participant, the impact factor $k_p$ and the contact duration $t_c$ as input. The latter two are determined in terms of the average pacing rate $\bar{f}_p$ (see Equations (3) and (4)). Before this model is used for the force reconstruction, it is analyzed how well the relations specified in Equations (3) and (4) fit the collected dataset. To do so, a single-step load pattern (Equation (2)) is fitted in a least-squares sense to each test for each participant, resulting in an optimal impact factor $k_p$ and contact duration $t_c$ in terms of the pacing rate $f_p$. The results in Figure 6 show that a good agreement is found with the relations specified in Equations (3) and (4). Furthermore, further analysis shows that fitting new relations between the impact factor, the contact time and the pacing rate, using only the data collected in this study, does not result in a better fit of the single-step load pattern as specified by Equation (2). This thus illustrates that the average values proposed by Equations (3) and (4) are also representative for the inter-person variability in the current dataset. Figure 7 compares the time series and amplitude spectrum of the measured and the reconstructed forces for participant 1 test 1. This figure shows that a very good agreement is found between the true and the reconstructed forces for both the time series and the frequency domain. The agreement is considerably better than the one obtained when applying Newton's second law (Figure 4).

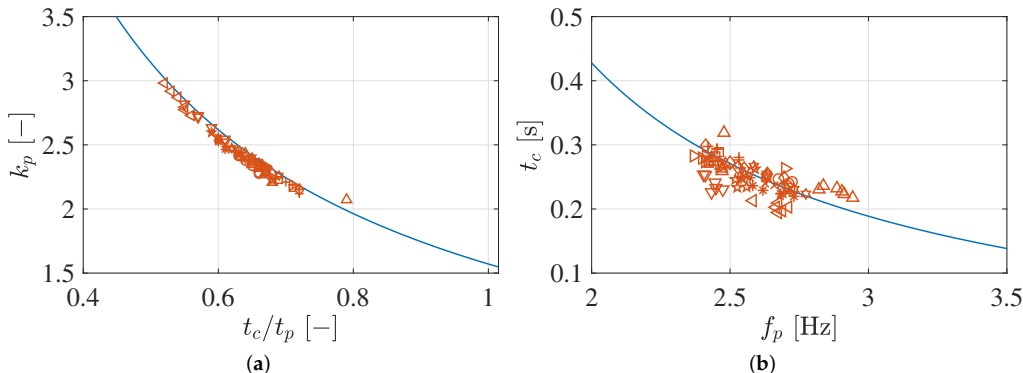

**Figure 6.** (**a**) The impact factor $k_p$ in terms of the ratio $t_c/t_p$: following Equation (4) (solid) and optimal fit for each test (symbol); (**b**) The contact time $t_c$ in terms of the pacing rate $f_p$: following Equation (3) (solid) and optimal fit for each test (symbol). A different symbol is used for each participant.

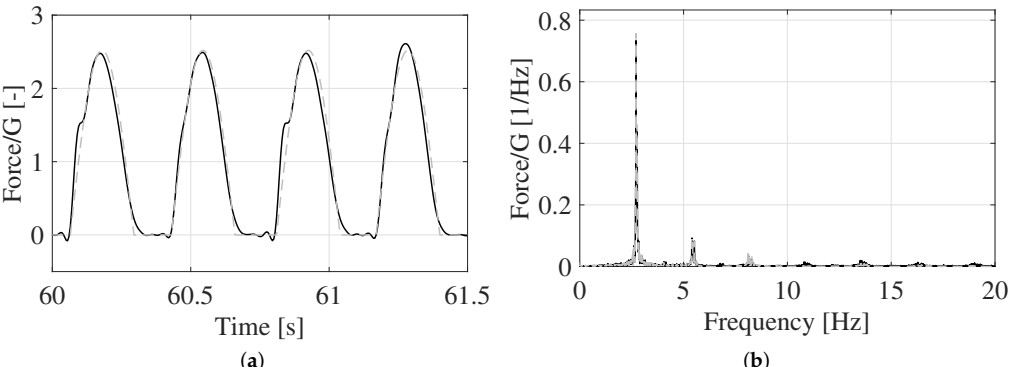

**Figure 7.** The load induced by participant 1 during test 1, measured forces (black), reconstructed using the method described in Section 2.2 and single-step load pattern model 1 (gray): (**a**) time series and (**b**) amplitude spectrum.

The last two load patterns (Models 2 and 3 in Table 1) only require the pacing rate as input.

The performance of the different reconstructions is now analyzed using $R_t$, $R_f$ and $R_h$ (Section 2.3). The results are summarized in Table 3 and visualized in Figure 8. The following observations are made:

- Model ref: The errors introduced by disregarding the step-by-step variations in the single-step load pattern or by small errors in the identified time-variant pacing rate are negligible in terms of $R_t^2$, $R_f^2$ and $R_h$. The small reconstruction errors $(1 - R_h)$ observed for the higher harmonics increase with the number of the harmonic, with an average of 11% ($h = 2$), 22% ($h = 3$) and 26% ($h = 4$).
- The performance of Models 1 and 2 is comparable: As a result of using a generalized single-step load pattern, the errors marginally increase in terms of $R_t^2$, $R_f^2$, $R_{h=1}$ and $R_{h=3}$. $R_{h=2}$ decreases from an average value of 89% (model ref) to 81% (model 1) and 77% (model 2). $R_{h=4}$ decreases from an average value of 74% (model ref) to 43% (model 1) and 42% (model 2).
- When applying the single-step load pattern in Model 3, the reconstruction error in terms of $R_t^2$ and $R_f^2$ increases by approximately 5% in comparison to the performance of Models 1 and 2. The reconstruction errors $(1 - R_h)$ on the fundamental, second and third harmonic are considerably larger, with an average $R_h$ of 18% ($h = 1$), 46% ($h = 2$) and 51% ($h = 3$). This indicates that the forefoot-strike load pattern is less representative for the collected dataset.
- The reconstruction method discussed in Section 2.2 considerably outperforms the reconstruction method using Newton's second law, even when a generalized single-step load pattern is used (Model 1 or 2): Considering $P_{50}$: $R_t^2 = 95\%$ vs. 52%, $R_f^2 = 98\%$ vs. 83%,

$R_{h=1} = 101\%$ vs. $80\%$, $R_{h=2} = 81\%$ vs. $63\%$, $R_{h=3} = 112\%$ vs. $34\%$ and $R_{h=4} = 43\%$ vs. $28\%$. This observation is also visually obvious when the corresponding histograms of the coefficients of determination are compared, e.g., by comparing Figure 5 and Figure 9.

In summary: When applying the reconstruction method discussed in Section 2.2, the small errors in the identified time-variant pacing rate or the disregarded step-by-step variations in the single-step load pattern only result in small errors in the reconstruction of the higher harmonics ($h \geq 2$) of the running load. The coefficients of determination indicate a nearly perfect fit with the true forces ($R_t^2 \approx 98\%$, $R_t^2 \lesssim 100\%$). When a generalized single-step load pattern (Models 1 or 2) is used, the correspondence with the true forces is still very good to excellent with coefficients of determination of approximately $95\%$ ($R_t^2$) and $98\%$ ($R_f^2$). The fundamental and second harmonic can in this way be reconstructed with a maximum error of $5\%$ ($12\%$) and $55\%$ ($69\%$) following the interval $P_{25} - P_{75}$ ($P_5 - P_{95}$). This method is thus best suited for the reconstruction of the running load focusing on the contribution of the fundamental harmonic. This is, by far, also the dominant harmonic component (see also Section 3.2.1). In case also the higher harmonics are of interest, a reconstruction method should be used that can also account for the inter- and intra-person variabilities in the single-step load pattern.

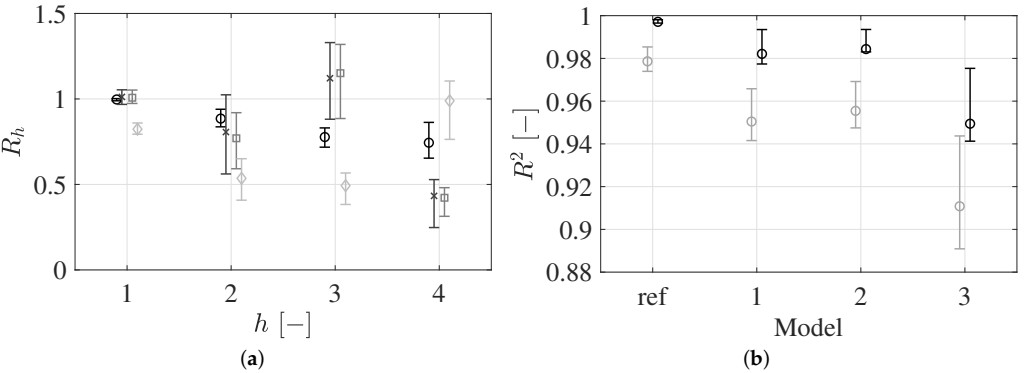

**(a)**                                                 **(b)**

**Figure 8.** (**a**) $R_h$ for the first four harmonics: mean value (symbol: model ref ($\circ$), model 1 ($\times$), model 2 ($\square$) and model 3 ($\diamond$)) and error bar indicating the ($P_{25} - P_{75}$) interval; (**b**) The coefficients of determination for the time domain ($R_t^2$, gray) and the frequency domain ($R_f^2$, black) for the different single-step load models.

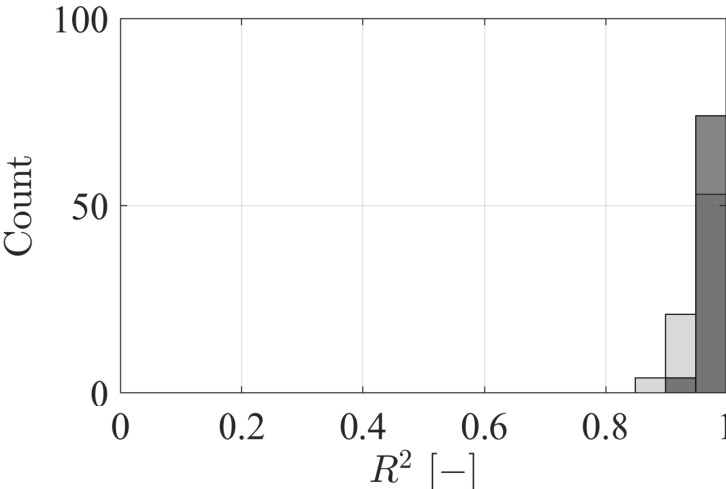

**Figure 9.** Histogram of the coefficients of determination for the time domain ($R_t^2$, light) and the frequency domain ($R_f^2$, dark) for the forces reconstructed using the method described in Section 2.2 and single-step load pattern in Model 1.

**Table 3.** The performance of the reconstruction methods.

| Model | $R_t^2$ (%) | | | $R_f^2$ (%) | | | $R_h(P_{50})$ (%) | | | |
|---|---|---|---|---|---|---|---|---|---|---|
| | $P_{50}$ | $P_{25}$ | $P_{75}$ | $P_{50}$ | $P_{25}$ | $P_{75}$ | $h = 1$ | $h = 2$ | $h = 3$ | $h = 4$ |
| Newton | 52 | 36 | 70 | 83 | 78 | 91 | 80 | 63 | 34 | 28 |
| ref | 98 | 97 | 99 | $\lesssim 100$ | $\lesssim 100$ | $\lesssim 100$ | $\lesssim 100$ | 89 | 78 | 74 |
| 1 | 95 | 94 | 97 | 98 | 98 | 99 | 101 | 81 | 112 | 43 |
| 2 | 96 | 95 | 97 | 98 | 98 | 99 | 101 | 77 | 115 | 42 |
| 3 | 91 | 89 | 94 | 95 | 94 | 98 | 82 | 54 | 49 | 99 |

### 3.3. Full-Scale Tests

The full-scale tests are performed on the Eeklo footbridge (Figure 10), which has a total length of 96 m, a main span of 42 m and two side spans of 27 m. Its cross-section consists of two main beams with a height of 1.2 m at a spacing of 3.4 m, supporting a steel deck of 8 mm thickness via three secondary beams. The structure is simply supported with land abutments at the sides and two piers at the mid span. The dynamic behavior of the Eeklo footbridge was identified in previous research [30], and a digital twin of the footbridge is available [52]. The experimentally identified mode shapes of the first six modes are visualized in Figure 11. For more information related to the footbridge and the model calibration, the reader is referred to [30]. The very low relative error $\varepsilon_j$ (<2%) and the high MAC values ($\lesssim 1$) show that there is an excellent agreement between the numerically predicted and the experimentally identified modal parameters for all modes with a natural frequency up to 12 Hz.

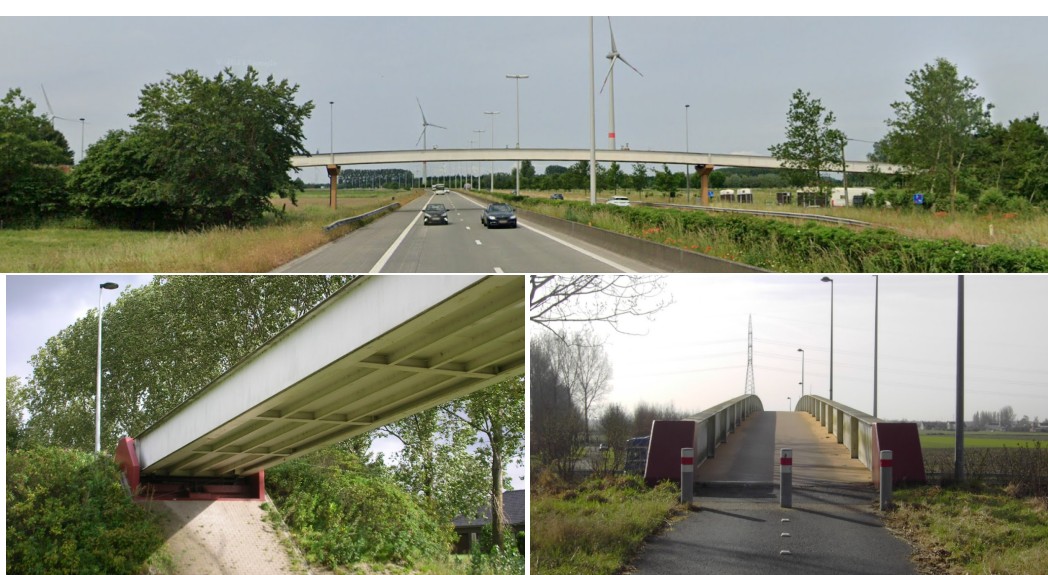

**Figure 10.** Eeklo footbridge.

The full-scale tests of interest in this work are tests involving running persons. During these tests, each participant was equipped with an inertial sensor (USB accelerometer X16-1D [50]). Using a pelvis belt, the sensors were securely fixed to the lower back of each person (and thus, close to their center of mass). The sampling rate of the devices was set to 200 Hz. The imperfect real running loads can then be reconstructed using the methodology discussed in Section 2.2. This section presents the results obtained using reconstruction Model 1. Similar results are obtained when Model 2 is applied, which is to be expected, as a comparable performance was found for Models 1 and 2 (Section 2.3). The vertical accelerations at the midspan of the bridge are registered using uni-axial acceleration sensors (PCB 393 B04). The data acquisition is performed using an NI 9234 4-Channel Sound and Vibration Input Module in a NI cDAQ-9171 CompactDAQ Chassis. The sample rate was set to 2048 Hz.

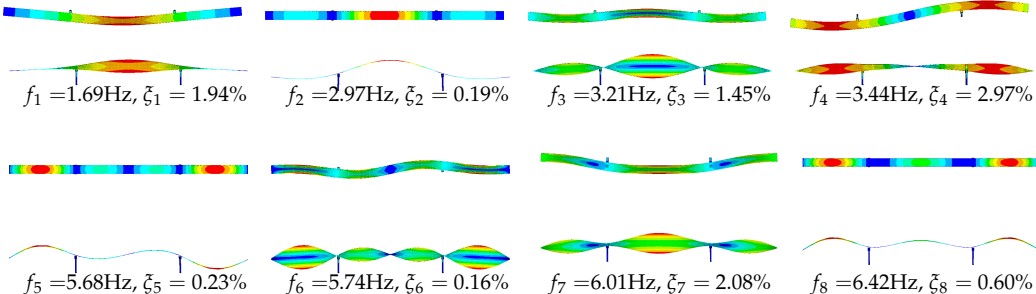

$f_1 = 1.69\text{Hz}, \xi_1 = 1.94\%$ $f_2 = 2.97\text{Hz}, \xi_2 = 0.19\%$ $f_3 = 3.21\text{Hz}, \xi_3 = 1.45\%$ $f_4 = 3.44\text{Hz}, \xi_4 = 2.97\%$

$f_5 = 5.68\text{Hz}, \xi_5 = 0.23\%$ $f_6 = 5.74\text{Hz}, \xi_6 = 0.16\%$ $f_7 = 6.01\text{Hz}, \xi_7 = 2.08\%$ $f_8 = 6.42\text{Hz}, \xi_8 = 0.60\%$

**Figure 11.** Top and side view of the first eight modes of the Eeklo footbridge (cfr. digital twin [52]).

The second mode of the Eeklo footbridge (a vertical bending mode) is expected to be prone to running-induced vibrations: it has a natural frequency ($\approx 3$ Hz) that falls within the interval of the dominant harmonic of the running load, and its modal mass and modal damping ratio ($\approx 0.2\%$) are low. Some tests will therefore target resonance with this mode.

Three types of tests involving running persons were performed on the Eeklo footbridge (see Table 4): (1) a single person running synchronized with mode 2, (2) three persons running one behind the other, synchronized with mode 2 and (3) seven persons running in a row (one behind the other) at a self-selected pacing rate. Each test type was repeated four times, in the following referred to as subtests. In case resonance was targeted with the second mode of the Eeklo footbridge, a metronome was used to tune the pacing rate of the runners. Each test type was repeated four times (four crossings per test type), which in the following are referred to as subtests. A subtest consists of a sequence of a build-up (during crossing) and a free decay in the structural response. Figure 12 presents the response for each test and subtest. Concerning the participants, the persons involved in test 1, 2 and 3 include respectively one, two, and three persons that were also involved in the laboratory experiments (Section 3.1).

For each test and subtest, the structural response is simulated using the digital twin of the footbridge and the reconstructed running loads. As the fundamental harmonic of the running load can in this way be reconstructed with a maximum error of 12% ($P_5 - P_{95}$, Section 3.2), the simulations are also performed considering the $P_5$ lower and $P_{95}$ upper bound of the expected structural response.

**Table 4.** Information on the tests performed on the Eeklo footbridge.

| Test | # Persons | Type | Trajectory |
|------|-----------|------|------------|
| 1 | 1 | target = synch with mode 2 | at 1/4th off the width of the cross section |
| 2 | 3 | target = synch with mode 2 | at 1/4th off the width of the cross section |
| 3 | 7 | self-selected pacing rate | along the centerline of the cross section |

Figure 13 presents the amplitude spectrum of the vertical midspan accelerations. This figure shows that the structural response is clearly dominated by the contribution of the second mode of the Eeklo footbridge ($f_2 \approx 3$ Hz). The same observation is made for the other tests.

Figure 12 presents the measured and simulated response for each test and subtest. The following observations are made:

- The structural response is highly similar for the four subtests in tests 1 and 2. This illustrates that the subtests are representative for the involved load case. More variation is observed among the subtests of test 3. This is due to the fact that for test 3, no metronome was used to tune the pacing rates: the individually self-selected running speeds are associated with a large degree of inter- and intra-person variability in terms of pacing rate. This is also illustrated by the distribution of pacing rates in tests 1, 2 and 3, as presented in Figure 14.

- Although the correlation between the simulations and the measurements is very high, in terms of amplitude, the simulations consistently overestimate the measured structural response, even when the lower bound of the simulations is considered. Considering $P_{50}$ ($P_5$), tests 1, 2 and 3, respectively, overestimate the structural response by 35% (19%), 47% (30%) and 31% (16%). When comparing test 1 and test 2, the degree to which the structural response is overestimated increases with the number of involved participants and/or with the increasing structural response. At first sight, this observation does not apply when test 1 (involving one participant) and 2 (involving three participants) are compared with test 3 (involving seven participants) where the highest number of participants is involved. The structural vibration levels are in this case comparable to these observed for test 2, yet the degree of overestimation is comparable to that observed for test 1. However, for test 3, no metronome signal was used to target resonance with mode 2, and this test type is therefore characterized by a different (and less resonant) distribution of pacing rates (see also Figure 14). In contrast, the distribution of pacing rates observed for test 1 and 2 is highly similar (as expected). These distributions explain why the structural response for test 3 is not higher than that of test 2. Furthermore, it is also expected that as a lower number of participants is running synchronized with the structural response, also the impact of HSI will be less, and in any sense, different from the impact observed in tests 1 and 2.

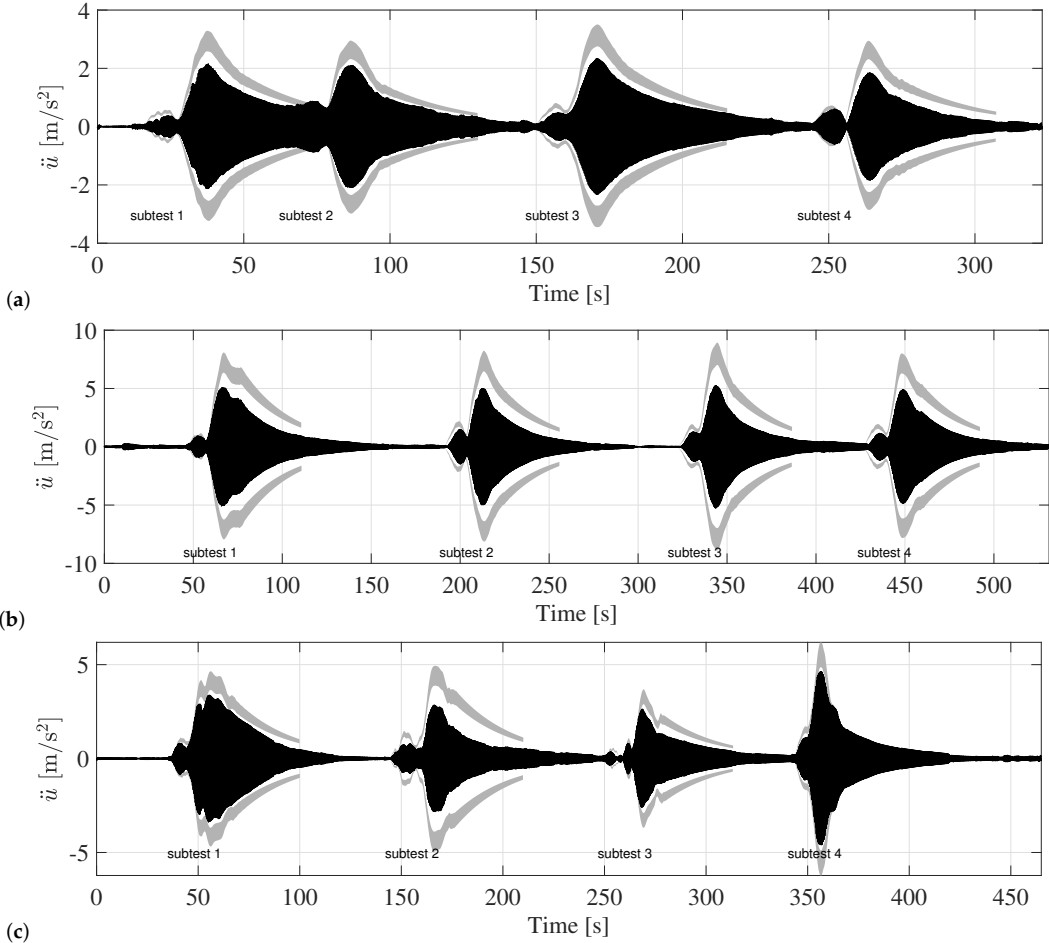

**Figure 12.** The structural response measured (black) and simulated (gray, lower $P_5$ and upper $P_{95}$ bound) at midspan during (**a**) test 1, (**b**) test 2 and (**c**) test 3.

The results indicate that the methodology used to reconstruct the running load, although suitable for the reconstruction on rigid surface, leads to an overestimation of the running load induced on flexible surfaces. Similar to walking-induced loading, these

results suggest that HSI phenomena are at play. These phenomena can involve added damping [24,52,53] or (un)conscious changes in the running behavior in response to the perceived vibrations [3,5,54]. With respect to the latter, it is noted that the runners did perceive the structural vibrations (in particular, during test 2 and 3), and some of them indicated that they had the impression that these perceived vibrations influenced the timing of their steps. To the knowledge of the authors, HSI phenomena for dynamic running actions have not been reported in the literature so far. The study of these HSI phenomena falls outside the scope of this work. This study, and in particular the study of active HSI, would furthermore greatly benefit from the analysis of the contact forces between the vibrating structure and the feet of the runner, which are not available for these experiments.

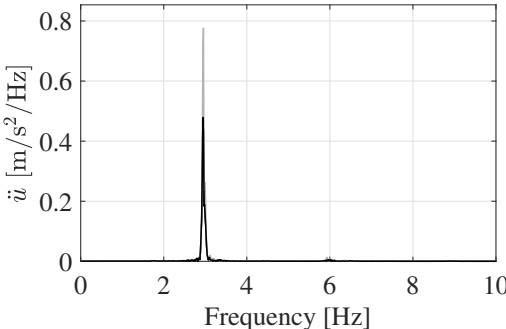

**Figure 13.** The amplitude spectrum of the vertical structural accelerations measured (black) and simulated (gray, $P_{50}$) at midspan during subtest 4 of test 1.

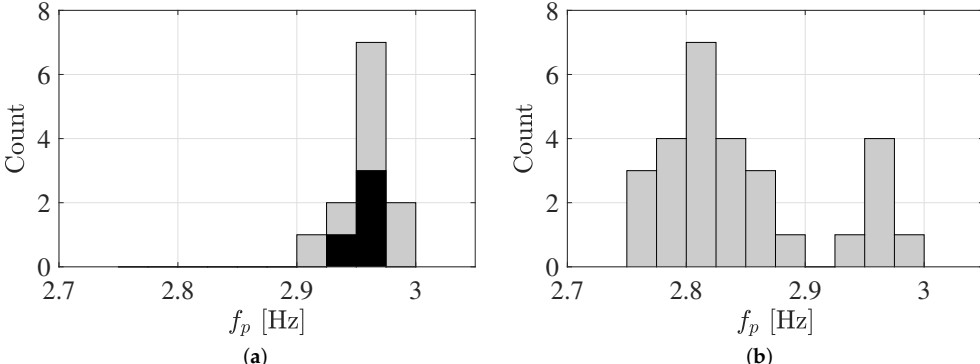

**Figure 14.** Histogram of the average pacing rate per participant for all subtests: (**a**) test 1 (black) and test 2 (gray) and (**b**) test 3.

## 4. Conclusions

A methodology is presented to reconstruct the running load based on a generalized single-step load pattern available in the literature and the time-variant pacing rate identified from the body accelerations. In this way, a good to excellent correspondence is found with the true forces registered on a rigid laboratory floor, with coefficients of determination of 95% in the time domain and 98% for the amplitude spectrum. It is furthermore shown that the fundamental and second harmonic can be reconstructed with a maximum error of 5% (12%) and 55% (69%) following the 25–75% (5–95%) percentile interval. This method is thus best suited for the reconstruction of the running load when the focus is on the contribution of the fundamental harmonic. This harmonic is, by far, the most dominant harmonic of the running load, representing nearly 90% of the energy concentrated in the first four harmonics. In case also the higher harmonics are of interest, a reconstruction method should be used that can also account for the inter- and intra-person variabilities in the single-step load pattern.

In a final step, the method is used to reconstruct the running load for in-field tests performed on a flexible footbridge sensitive to human-induced vibrations. The tests involved

a single runner, three runners or seven runners, each of them equipped with a wireless inertial sensor attached close to their center of mass. The reconstructed load and a digital twin of the footbridge are then used to simulate the structural response. The correlation between the simulated and the measured structural response is very high. In terms of amplitude, however, the simulations consistently overestimate the measured structural response (up to 50%). Even when the 5th percentile lower bound of the simulations is considered, the structural response is easily overestimated by 15% to 30%. The results suggest that HSI phenomena such as added damping or the (un)conscious changes in the running behavior in response to the perceived vibrations are at play. Further research is necessary to investigate HSI for dynamic running actions.

**Author Contributions:** Conceptualization, K.V.N.; methodology, K.V.N.; writing—original draft preparation, K.V.N.; writing—review and editing, B.V. and P.V.d.B.; funding acquisition, K.V.N. All authors have read and agreed to the published version of the manuscript.

**Funding:** The first author is a postdoctoral fellow of the Research Foundation Flanders (FWO, application number 12E0819N). The financial support is gratefully acknowledged.

**Institutional Review Board Statement:** The study was conducted in accordance with the Declaration of Helsinki, and approved by the Institutional Review Board (or Ethics Committee) of KU Leuven (G-201703810 and 22 March 2017).

**Informed Consent Statement:** Informed consent was obtained from all subjects involved in the study.

**Data Availability Statement:** The data presented in this study are openly available in KU Leuven Research Data Repository (RDR) at https://doi.org/10.48804/UGL5RS.

**Conflicts of Interest:** The authors declare no conflict of interest.

## Abbreviations

The following abbreviations are used in this manuscript:

| | |
|---|---|
| BCoM | Body center of mass |
| GRFs | Ground reaction forces |
| HHI | Human–human interaction |
| HSI | Human–structure interaction |
| PSD | Power spectral density |
| TMD | Tuned mass damper |

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
