# Peer review of "Reconstruction of the Vertical Dynamic Running Load from the Registered Body Motion"

_vibration, doi:10.3390/vibration5030026_

Round 1
Reviewer 1 Report
It is not clear how the single-step load patterns are arranged to obtain the average load pattern. From Figure 3, it seems that the ends of the individual patterns coincide at the same instant of time. Please, clarify.
Reviewer 2 Report
The study presented in the paper focuses on dynamic running-induced loads and proposes a methodology to reconstruct the vertical running load from registered body motion. The reconstructed forces are subsequently used to numerically simulate the dynamic response of the Eeklo footbridge under the action of running pedestrians and comparison with acceleration recorded during in-field tests is performed.
The topic treated in this study is of great interest for the scientific community dealing with human-induced vibrations, since very little research has been performed so far on running excitation and codes of practice lack of reliable load models.
The paper is well written and the study clearly presented. In my opinion the paper can be accepted for publication in the present form, apart from some minor issues in Section 3.3:
- In this section measured and simulated responses are compared. It is not clear which method of reconstruction has been adopted for the forces applied to the footbridge model among the four presented in the previous section. According to the comments reported at the end of page 11, I suppose that Model 1 or 2 has been adopted, but it should be clearly stated;
- Figure 13: for each test a single acceleration time history is reported for the measured response. Is it the average response obtained from the 4 subtests for each test?
